# A Study on the Effect of Doping Metallic Nanoparticles on Fracture Properties of Polylactic Acid Nanofibres via Molecular Dynamics Simulation

**DOI:** 10.3390/nano13060989

**Published:** 2023-03-09

**Authors:** Razie Izadi, Patrizia Trovalusci, Nicholas Fantuzzi

**Affiliations:** 1Department of Structural and Geotechnical Engineering, Sapienza University of Rome, Via Gramsci 53, 00197 Rome, Italy; razie.izadi@uniroma1.it; 2Department of Civil, Chemical, Environmental and Materials Engineering, University of Bologna, Viale del Risorgimento 2, 40136 Bologna, Italy; nicholas.fantuzzi@unibo.it

**Keywords:** molecular dynamics simulation, fracture properties, doped polymer nanofibre, doping metallic nanoparticles, biodegradable poly(lactic acid), mechanical properties, material characterization

## Abstract

All-atom molecular dynamics simulations are conducted to elucidate the fracture mechanism of polylactic acid nanofibres doped with metallic nanoparticles. Extensional deformation is applied on polymer nanofibres decorated with spherical silver nanoparticles on the surface layer. In the obtained stress–strain curve, the elastic, yield, strain softening and fracture regions are recognized, where mechanical parameters are evaluated by tracking the stress, strain energy and geometrical evolutions. The energy release rate during crack propagation, which is a crucial factor in fracture mechanics, is calculated. The results show that the presence of doping nanoparticles improves the fracture properties of the polymer nanofibre consistently with experimental observation. The nanoparticles bind together polymer chains on the surface layer, which hinders crack initiation and propagation. The effect of the distribution of nanoparticles is studied through different doping decorations. Additionally, a discussion on the variation of internal energy components during uniaxial tensile loading is provided to unravel the deformation mechanism of nanoparticle-doped nanofibres.

## 1. Introduction

Polymer nanofibres (PNFs), with diameters less than 100 nanometres (nm), have gained increasing attention due to their innovative application in diverse areas of science. Due to efficient production techniques such as electro-spinning, polymer nanofibres can be produced with versatility and low cost [1,2,3,4,5], enabling their effective employment in different sectors including tissue engineering [6,7,8,9,10,11] drug delivery and wound dressing [9], reinforcements in nano-composites [12,13], energy storage devices [14], sensors [15,16] and packaging [17]. Recently, the use of compostable and biodegradable polymers is motivated due to the growing concerns over environmental pollution and sustainability [18]. However, compostable polymers still present some limitations compared to the traditional petroleum-based ones, such as intrinsic brittleness, low impact resistance and toughness, poor thermal and gas barrier properties. In order to overcome these drawbacks, a solution is to dope polymer nanofibres with nanofillers such as metallic nanoparticles to enhance mechanical, electrical, and thermal properties [19,20,21]. 

Experiments have shown that the addition of well-dispersed metallic nanoparticles to polymer nanofibres can improve their mechanical properties such as tensile strength, Young’s modulus, and toughness [22]. The size, shape, and distribution of the metallic nanoparticles also play an important role in the mechanical properties of the final material [19,20,22]. 

Cacciotti et al. [19] conducted an experiment on PLA nanofibres mat doped with silver nanoparticles and reported an enhancement of around 25% in the mechanical strength of the resulting material.

In an experimental work, Efome et al. [21] studied metal–organic nanoparticles supported on polyacrylonitrile (PAN) nanofibre membrane and claimed that the tensile mechanical properties, namely, Young’s modulus, yield stress, elongation at break, and stress at break of the doped fibrous membrane are superior to the neat one attributed to the strong interaction between PAN and nanoparticles. Wu et al. [23] studied nanoscale aluminium oxide whisker-doped Poly(ε-caprolactone) nanofibres and reported a significant increase in the elastic modulus of nanofibres by the well-aligned aluminium oxide.

In addition to the enhancement of mechanical properties, the metallic nanoparticles improve the capacitance and energy density of the polymer nanofibres which is applicable in supercapacitors [14]. The improved electrical and thermal conductivity of polymer nanofibres doped with metallic nanoparticles (PNF/MNP) promotes their sensing applications, such as temperature, humidity, and gas sensing [20]. 

From the manufacturing aspect, besides electrospinning which allows for the uniform dispersion of nanoparticles within the polymer fibres, other fabrication methods such as solvent casting, self-assembly, and template-assisted can also be adopted to produce PNF/MNP. 

The type of polymer and metallic nanoparticles also affects the mechanical properties of the resulting material. Poly(lactic acid) (PLA) is a biodegradable and compostable polymer ideal for the production of high-quality electro-spun nanofibres due to its high molecular weight and low melting temperature [11,24,25]. Besides its excellent mechanical properties and cutting-edge applications in nano-composites [26], the biodegradability and biocompatibility of PLA promote its biomedical usage where it can be safely absorbed by the body after serving its purpose [10,11,24,25,27]. In addition to PLA, polyvinyl alcohol (PVA) and polyvinylidene fluoride (PVDF) are also commonly used polymers in the fabrication of PNF/MNP. On the other hand, gold, silver, copper, and metal oxides such as titanium oxide or iron oxide and even particles comprised of metal–organic framework (MOF) compounds are commonly used as metallic nanoparticles [4,21,28]. Silver nanoparticle is a popular choice in biomedical applications due to their antibacterial and antiviral characteristics. By doping nanofibres with silver nanoparticles, the energy harvesting is also significantly improved due to the surface plasmon resonance effect of silver nanoparticles and the mechanical flexibility of nanofibres [29].

Understanding the mechanical behaviour of nanofibres helps in optimizing their performance, ensuring their durability and reliability in real-world scenarios. For instance, the stiffness of PNF artificial substrates influences cell growth, proliferation, and differentiation in tissue engineering. In addition, the strength and durability of PNF are critical factors in the fields of filtration, protective clothing, and fabrics [30,31,32]. On the other hand, the study of fracture mechanisms provides insight into the stability and longevity of nanofibres under different loading conditions.

Right now, the experimental methods to evaluate PNFs’ mechanical properties include atomic force microscopy (AFM), three-point bending, nanoindentation, and nano-tensile and resonance contact-based test [33]. These tests can be used for fibres with diameters larger than 20 nm [33]; however, for smaller diameters, it becomes challenging to handle and manipulate these fibres without altering their properties, leading to inaccurate results. For instance, in AFM cantilevers, anchoring PNF on micro ridges to prevent slippage (Baker et al., 2016) and applying the nano-force in the proper position [5,34] is quite difficult. Moreover, in nano-tensile tests, mounting single fibres to a conventional tensile tester and providing loadcells with sufficient sensitivity is challenging [33]. 

To address these challenges, atomistic simulations have become an important tool in the study of nanoscopic mechanical properties. Atomistic simulations allow for a detailed and precise investigation of the mechanical behaviour at the atomic scale, providing insight into the underlying mechanisms that govern the mechanical properties of nanofibres. Moreover, simulations can be performed under controlled conditions, eliminating the effects of external factors that may impact experimental results [35,36]. Molecular dynamics (MD) is one of the widely used atomistic simulation techniques that use classical mechanics to model the motion of atoms and molecules in a material [37]. In MD simulations, the interactions between the atoms and molecules are calculated using Newton’s laws of motion, allowing for close observation of atomistic phenomena that govern the gross mechanical behaviour [38,39]. MD also enables observing and analysing the deformation mechanisms, such as slip, twinning, and fracture, that occur at the atomic scale. 

Until now, a limited number of studies have employed atomistic simulations to study the mechanical properties of PNFs; Vao-soongnern et al. [40] were the first to use atomistic simulations to model polyethylene (PE) nanofibres, calculating surface energies as a function of fibre radius on a high coordination lattice using the coarse-grained Monte Carlo approach. Curgul et al. [41] later used coarse-grained MD and united atom model to study the size-dependent glass transition temperature of PE nanofibres, while Buell et al. [42] adopted MD to calculate Young’s modulus and Poisson’s ratio of glassy PE nanofibres. Milani et al. [43] simulated nylon-6 nanofibres and studied their melting and re-crystallization using a coarse-grained MARTINI force field [44]. Tang et al. [45] discovered Young’s modulus and Poisson’s ratio of a collection of bead–spring chains representing PNF through MD simulation. Deng et al. [46] investigated the effect of pre-stretching on the elastic modulus of PNFs through MD simulations. Later, Peng et al. [47] presented the stress–strain curves and Young’s moduli for hot-drawn PE nanofibres with diameters of 10 nm to 19 nm. In a research by Kwon and Sung [48], the nonlinear mechanical response of the nanofibre and its dependency on strain rate were examined using a coarse-grained MD on a typical PNF with a 30 nm diameter. Finally, in a recent publication by Liu et al. [49], the tensile behaviour of a special type of nanofibre known as a hierarchically twisted nanofibre was explored by coarse-grained MD, and the impact of twisting angle was discussed. 

Considering the above remarks, a study based on atomistic simulations is favourable to enlighten the effect of doping nanoparticles on the mechanical and fracture properties of polymer nanofibres. To the best of our knowledge, this study is not reported in the literature, therefore, in the current work, an all-atom MD simulation is conducted to study the mechanical response of PLA nanofibres doped with silver nanoparticles. Special attention is devoted to the fracture of nanofibres and the effect of doping nanoparticles on the underlying fracture mechanism. More precisely, extensional deformation is applied on polymer nanofibres decorated with spherical silver nanoparticles on the surface layer where mechanical parameters including elastic, plastic and fracture measure, are evaluated by tracking the stress, strain energy and geometrical evolutions. In addition, by examining different decorations, the effect of nanoparticle distribution is studied. The internal energy components during the loading are being studied in an effort to learn more about the deformation mechanism.

The rest of the paper is structured as follows: Section 2 describes the overall simulation methodology. The obtained results are presented in Section 3 followed by a comprehensive discussion in Section 4. Conclusive remarks are reported in Section 5. 

## 2. Materials and Methods

### 2.1. Materials

In the current work, molecular dynamics simulations are conducted using LAMMPS [50] which is an efficient tool for simulating large-scale atomistic systems and enables spatial decomposition of the domain on parallel processors. Rendering and a part of post-processing are performed through the Open Visualization Tool (OVITO, version 3.7.9, OVITO GmbH, Germany) developed by Stukowski et al. [51] while the initial structures are created in Materials Studio software [52]. Nanofibres are composed of PLA which is a biodegradable polymer obtained from renewable sources and agricultural products with very low toxicity [53]. The molecular formula of PLA is ${\left(C_{3}H_{4}O_{2}\right)}_{n}$ [54], which is shown in Figure 1. Silver nanoparticles (Ag-NP) with a diameter of 1.8 nm are used as the doping agent of PLA nanofibre, as shown in Figure 2a. Polymer nanofibres (PNFs) with a diameter of 5.4 nm are constructed from 90 numbers of 20-monomer polymer chains (Figure 2b), where each monomer contains three carbon atoms, four hydrogen atoms, and two oxygen atoms (Figure 2c). 

### 2.2. Initial Configurations

Prior to creating nanofibres, a cuboid periodic simulation box is constructed from 90 numbers of the polymer chains using an Amorphous Cell Module in Materials Studio (BROVIA Materials Studio 2022, Dassault Systèmes) [52] that build molecules in a cell in a Monte Carlo manner by minimizing close contacts between atoms. To minimize high energy interactions, this representative volume element (RVE) is initially built at a low density (0.3 g/cm^3^) and is subsequently ramped up to an estimated density of 1.3 g/cm^3^, in accordance with the literature [55]. 

Then, energy reduction is conducted by a cascade of the steepest descent, modified basis Newton–Raphson, and quasi-Newton technique to assure an optimum starting structure. The RVE temperature is then raised and maintained at 500 K using a canonical (NVT) ensemble, which is expected to be above the glass transition temperature (Tg) of PLA (between 323 K–353 K [56]) to facilitate mobility of polymer chains and speed up equilibration [57]. An NPT isothermal–isobaric ensemble is then followed at 1 atm and 500 K for 5 nanoseconds (ns). The system is finally cooled to 298 K in 10 nanoseconds under constant pressure, reaching its ultimate dimension of 4.8 × 4.8 × 7.62 nm^3^ (Figure 3a). The resultant average density of 1.237 g/cm^3^ matches well with available data (between 1.21–1.3 [55]). At this stage, the Nose–Hoover thermostat and barostat are adopted with a time step of 1 femtosecond (fs) using a stochastic time integrator [58]. 

To create PLA nanofibres, after the cuboid simulation box has reached equilibrium, without rescaling atom coordinates, the x and y dimensions of the box are extended by 5–7 times the cut-off radius (1 nm) while the z-direction dimension is left unaltered. In this way, we can construct a nanofibre isolated in the transverse direction with an infinite axial length while keeping the simulation box’s periodicity. 

The silver nanoparticles are manually positioned on the surface of the nanofibres, followed by optimization of energy and an NVT ensemble of 50 ps to ensure optimal positioning and creation of the intermolecular network between the nanofibre and the nanoparticle (Figure 3c). In Figure 3b some ripplings on the surface of the nanofibre are observed which is consistent with available SEM images [45]. The resulting structure is then imported into LAMMPS using the msi2lmp plugin [50]. The Langevin thermostat [59] within the microcanonical ensemble (NVE) is used during deformation. Since the lateral size of the simulation box is significantly larger than the cross section of the nanofibre, the lateral surface of the fibre has unconstrained boundary conditions and is free to expand or contract, similar to a uniaxial tensile test. The velocity–Verlet algorithm integrates Newton’s equations of motion with a time step of 0.5 fs. 

In the current work, we consider pristine PLA nanofibre, a nanofibre doped with 2 and 3 silver nanoparticles within the simulation box which correspond to 3.9% and 5.8% weight fraction of doping agent to the nanofibre. This weight fraction is consistent with the experimental work on metallic doped polymer nanofibres in the context of nanofibrous networks [19,20,21,22,23]. For instance, Cacciotti et al. [19] studied the effect of 1% weight fraction of Ag nanoparticle on the mechanical properties of ternary fibrous mats based on PLA.

### 2.3. Interatomic Potential

In the current work, all-atom simulations are implemented instead of the coarse-grained approach to increase the precision and reliability of the results. This is because in the coarse-grained method, a group of atoms are treated as a single bead and internal interactions are ignored, which lowers the computational cost while decreasing the precision. The ab initio Class II polymer consistent forcefield (PCFF) is assigned to represent atomic interactions since it is adept at modelling polymers and organic materials [60,61]. PCFF is well-known for its ability to characterize cohesive energies, mechanical characteristics, compressibility, heat capacities, and elastic constants [62]. The total energy in PCFF (EPCFF) is divided into valence\bonded (Evalence) and non-bonded interactions (Enon−bonded) according to Equations (1)–(4):(1)EPCFF=Evalence+Enon−bonded,
where the valence term covers the bond (Ebond), angle (Eangle), dihedral (Edihedral), inversion (Einversion ), and the cross-coupling (Ecross−coupling) interactions between atoms: (2)Evalence=Ebond+Eangle+Edihedral+Einversion +Ecross−coupling.

Bond and angle interactions include up to quartic terms to describe inharmonic properties. For torsion interactions, a symmetric Fourier expansion is utilized, with a simple harmonic function representing the out-of-plane contributions [63].
(3)$$\begin{array}{l} E_{\mathrm{bond}}=\sum_{n=2}^{4}k_{n}^{b}{\left(b-b_{0}\right)}^{n},\;E_{\mathrm{angle}}=\sum_{n=2}^{4}k_{n}^{\theta }{\left(\theta -\theta_{0}\right)}^{n},\\ E_{\mathrm{dihedral}}=\sum_{n=2}^{4}k_{n}^{\phi }{\left(1-\cos\left(n\phi \right)\right)}^{n},\;E_{\mathrm{inversion}}=k^{\chi }{\left(\chi -\chi_{0}\right)}^{2}, \end{array}$$
where $k_{i}^{j}$ represents i-th coefficient related to j-th kind of energy where the superscripts b, θ, ϕ and χ are assigned for bond, angle, dihedral and inversion. More information on the cross-coupling term can be found in [63].

The non-bonded term includes electrostatic ($E_{\mathrm{electrostatic}}$) interactions and a 9–6 Lennard–Jones potential to account for the van der Waals forces ($E_{\mathrm{van}\;\mathrm{der}\;\mathrm{Waals}}$).
(4)$$\begin{array}{l} E_{\mathrm{non}-\mathrm{bonded}}=E_{\mathrm{electrostatic}}+E_{\mathrm{van}\;\mathrm{der}\;\mathrm{Waals}},\\ E_{\mathrm{van}\;\mathrm{der}\;\mathrm{Waals}}=\epsilon \left[{\left(\frac{\sigma }{r}\right)}^{9}-{\left(\frac{\sigma }{r}\right)}^{6}\right]\;r<r_{\mathrm{cutoff}},\\ E_{\mathrm{electrostatic}}=q_{i}q_{j}/r, \end{array}$$
where r is the distance between an atom pair, ϵ and σ are the well depth and zero-crossing distance and $q_{i}$ is the charge of atom i. For all simulations, the cut-off radius is r_cutoff_ = 1 nm [64].

The cut-off radius for all simulations is r_cutoff_ = 1.05 nm. Extending the cut-off radius further raises simulation computational cost dramatically, as non-bonded interactions are the most time- and computation-consuming elements of MD simulation. We have avoided adding this computational overhead, since the capability of using 1 nm to recreate PLA parameters is already addressed in the literature [64,65,66]. In addition, the use of all-atom MD simulation in the current work has enhanced simulation accuracy at the expense of higher computational cost as compared to previous work on coarse-grained MD modeling of PLA (for instance, [67]).

### 2.4. Deformation Simulation

After equilibration of pristine and silver-doped nanofibres, as shown in Figure 4, a uniaxial deformation with a constant strain rate is applied on each nanofibre by evenly increasing the box size in the fibre’s axial direction. Since the lateral surface is unconstrained, the deformation is analogous to a uniaxial tensile test. 

The Virial theorem is used to determine the stress tensors as shown in Equation (5):(5)$$\sigma_{\mathrm{ij}}=\frac{-1}{V}\underset{\alpha }{\sum }(m^{\alpha }v_{i}^{\alpha }v_{j}^{\alpha }+\frac{1}{2}\underset{\beta \neq \alpha }{\sum }r_{i}^{\alpha \beta }F_{j}^{\alpha \beta }),$$
where i and j show coordinate axes, V is the simulation box volume, $r_{i}^{\alpha \beta }$ is the ith element of the distance vector of atom α to β and $F_{j}^{\alpha \beta }$ is the jth element of the force exerted from atom β on α. 

Through the stress–strain curve, Young’s modulus is determined using the slope of the linear elastic regime (Figure 5a) and the Poisson’s ratio is obtained by monitoring the radial deformation in the same regime. The proportional and yield stresses are also determined as well as the post-yield parameters such as toughness and fracture properties. To reduce the noise on the stress data, a regression with the smooth function in MATLAB using a moving average filter is applied.

### 2.5. Energy Release Rate

Energy release rate (G) is a key concept in fracture mechanics that measures the amount of energy required to increase the size of a crack in a material. It is defined as the rate at which energy is released as a crack extends, and is calculated using the difference between the elastic energy stored in the material and the energy required to create new surfaces as the crack extends. The energy release rate is an important factor in determining the stability of a crack and its propagation behaviour and is often used to evaluate the critical stress intensity factor, which determines the load at which a crack will propagate [68].

If $W_{ext}$ denotes the external work, U the stored strain energy and $A_{crack}$ the cracked area, the energy release rate (G) can be calculated using Equation (6) [68]: (6)$$G=-\frac{\Delta \left(U-W_{ext}\right)}{\Delta A_{crack}}.$$

In the current study,$\;W_{ext}$ is calculated as $\int_{0}^{l}\;Fdl$ through the integration of an interpolation function for force (F)-displacement (l) data with the help of “griddedInterpolant” function in MATLAB. The strain energy is obtained by the instant potential energy subtracted by its initial value. In Equation (6), l is the length of the simulation box in the z direction and dl is calculated as ϵ×l at each strain (ϵ) and the axial force (F) is obtained using the normal stress in the z direction ($\sigma_{zz}$).

### 2.6. Effective Radius ($R_{eff}$)

To calculate the Virial stress in Equation (5), the volume occupied by the system must be known; however, in LAMMPS, the volume in the calculation of Virial stress is the volume of the simulation box; hence, the obtained stresses must be rectified by the simulation box to fibre volume ratio.

In previous works on MD simulation of nanofibres, usually, the Gibbs dividing surface (GDS) has been employed to calculate the equivalent radius [41,42,46,69]. In a simplified expression, GDS for nanofibres determines the radius of a homogenous fibre with a density equal to that of the fibre core and with the same integral mass as the original fibre. However, in this work, we use a different approach which results in less noise during uniaxial deformation as compared to GDS. However, in this work, we employ a new technique that results in less noise during uniaxial deformation as compared to GDS. At each time step, the average radial distance ($R_{i}$) of all atoms from the centroid is set equal to the mean radius ($R_{\mathrm{ave}}$) of an equivalent solid circle. The radius of this equivalent cylinder is supposed as the effective radius ($R_{\mathrm{eff}}$) of nanofibres considering $R_{\mathrm{eff}}=3R_{\mathrm{ave}}/2$ (as illustrated in Figure 6).

## 3. Results 

### 3.1. Stress and Energy Curve

Figure 5a shows the isothermal stress–strain curve for silver-doped PLA nanofibre from uniaxial deformation. The curve has four regimes: elastic, plastic, strain softening and fracture. In the beginning, the stress increases linearly in proportion to the applied strain, indicating an elastic regime (relevant to Figure 4a). As the strain increases, the stress–strain slope declines until the curve flattens out, signifying the yield point. Upon this point, the material enters a strain-softening phase due to the slippage process of polymer chains (Figure 4b).

With further strain, the polymer chains begin to pull out slightly of each other signified as crack initiation (Figure 4c), accompanied by a sharp drop in the stress diagram as shown in Figure 5a.

By further loading, the polymer chains continue to pull out until the complete rapture occurs (Figure 4d,e). Note that an initial stress exists in the undeformed nanofibres where the strain magnitude is zero, which is in line with previous studies and is induced by nanofibre surface tension [42,48]. The initial stress is subtracted in Figure 5a.

The Young’s modulus, proportional and yield stresses are determined from the stress–strain curve (Figure 5a) for pristine and silver-doped PLA nanofibres, and presented in Table 1.

The increase in Young’s modulus and yield stress is in line with experimental observations of doped nanofibrous networks [19,20,21,22,23]. Furthermore, as illustrated in Figure 7, using the radial deformation in the elastic regime, Poisson’s ratio (ν) is determined for the three nanofibres and provided in Table 1.

In addition to the stress curve, the strain energy density ($\frac{\mathrm{Potential}\;\mathrm{energy}-\mathrm{initial}\;\mathrm{potential}\;\mathrm{energy}}{\mathrm{nanofibre}\;\mathrm{volume}}$) is tracked during deformation (Figure 5b). The strain energy density at the onset of yield and fracture is used to calculate the moduli of resilience and toughness, which measure the amount of strain energy density that a material can absorb before yielding or fracture. 

It should be highlighted that, as demonstrated in Figure 4 and Figure 5, nanofibres can withstand considerable deformations prior to full rapture. Experiments have consistently shown that nanofibres can sustain substantial strains due to several relaxation mechanisms, such as chain unfolding, molecular disentanglement and side group rotation. For example, in an experimental study by Naraghi et al. [70], individual electrospun polyacrylonitrile nanofibres (PAN) are stretched beyond 200% in a uniaxial nano-tensile test.

### 3.2. Energy Release Rate

As shown in Figure 5b, before crack initiation, all the work conducted on the nanofibre is stored as the strain energy; however, where the polymer chains begin to pull out, which we refer to as crack nucleation, a growing difference between $W_{ext}$ and U arises, which is associated with the energy release due to crack propagation.

The method to determine the energy release rate in the current work is illustrated in Figure 8. First, the cross-sectional area is calculated right before crack initiation $\left(A_{c}\right)$. Then, we consider the crack area equal to zero at crack initiation and equal to $A_{c}$ at the total separation; the energy release rate will be the difference of $U-W_{ext}$ at these two points divided by $A_{c}$. The determined values for G for pristine and doped nanofibres are listed in Table 1. 

## 4. Discussion

In this Section, the effect of doping silver nanoparticles on the elastic, plastic and fracture properties of nanofibres is studied. A discussion on radial density profile and evolution of energy contributions is also provided to gain a better insight into the structural properties and deformation mechanism.

### 4.1. The Effect of Doping Ag NP on Elastic and Plastic Properties of Nanofibres

In Figure 9, the distribution of atomic strain is compared for pristine and silver-doped nanofibres at 5% strain where nanofibres are in the linear elastic phase according to the stress curve in Figure 5a. As demonstrated in Figure 9, nanoparticles have little influence on the atomic strain distribution in the elastic phase, even though some localised strain arises in the vicinity of nanoparticles. According to the elastic and plastic parameters in Table 1, nanoparticles moderately enhance the brittleness of nanofibres, resulting in a greater Young’s modulus and yield stress, whereas yielding occurs slightly earlier in the doped nanofibres. In addition, the moduli of resilience and toughness have reduced in doped nanofibres, also indicating increased brittleness. These moduli represent the amount of strain energy density that a material can accommodate before yield or fracture, respectively.

As indicated in Table 1 and Figure 7, the Poisson’s ratio is decreased in doped nanofibres compared to the pristine one. This suggests that nanoparticles restrain the movement of the polymer chains and thus the deformability of nanofibres. In addition, this phenomenon also helps to avoid the occurrence of local necking as shown in Figure 10.

### 4.2. The Effect of Doping Ag NP on Fracture Properties of Nanofibres

Figure 10 compares the doped and untreated nanofibres at the strain of 85%. The colours show the atomic strain within the fibre in the range from 0 to 100%. It can be clearly seen that the integrity of the nanofibre is maintained by the presence of nanoparticles, whereas the pristine nanofibre is completely raptured at this strain. The nanoparticles also reduced the local atomic strain within the fibre, thus reducing the shrinkage of the cross-section due to the applied strain and preventing local necking. The nanoparticles act as consolidators, binding the polymer chain together and preventing it from pulling out. In addition, as provided in Table 1, the doping nanoparticles increase the critical energy release rate and thus hinder crack propagation.

### 4.3. The Effect of Doping Ag NP Weight Fraction 

To study the effect of the weight fraction of the nanoparticles, Figure 11 compares the distribution of atomic strain in pristine and doped nanofibres with 3.9% and 5.8% weight fractions of Ag NP. Although according to Table 1, the elastic properties of nanofibres doped with 3.9% of Ag NP are slightly higher than those doped with 5.8%, which may be due to the localised strain near the added nanoparticle, the improvement in fracture resistance is much more pronounced in the case of 5.8%. To facilitate discussion, the nanoparticles are numbered in Figure 11. As can be seen in this figure, the presence of NP numbered 3 at 5.8% weight fraction is quite effective in preventing polymer chain detachment and increasing fracture strength; however, it should be noted that the results also depend on the position at which the nanoparticles are launched. It can be seen from Figure 11 that the nanoparticle numbered 2 in the middle of the doped nanofibre did not play a favourable role as it attracted the polymer chains on one side of the initiated crack.

Although we did not detect a direct correlation between the weight percentage of the nanoparticles and the improved fracture properties, our results show that the proper distribution of nanoparticles can be very effective in increasing the fracture strength of nanofibres.

### 4.4. Density Radial Distribution

In Figure 12, the density distribution along the radius is studied for a better understanding of the internal structure of nanofibres. To determine the density profile along the radius, nanofibre cross-section is divided into concentric cylindrical layers of 0.1 nm thickness (schematically shown in Figure 12a). The mass density of each layer is obtained by counting the number of each atom type that has fallen into the particular shell and normalizing it by its volume.

The obtained density radial distribution indicates the presence of a core area with a density approaching the bulk state (1.24 g/cm^3^) encompassed by a degrading low-density layer known as the “surface layer” which is believed to be induced by nanofibre surface tension and affect the mechanical behaviour of the nanofibres [41,47]. The density fluctuates near the fibre centre, as reported in earlier research [41,42,47] due to weaker statistical sampling for shells with a small radius. From Figure 12b, it seems that the presence of doping nanoparticles on the surface of nanofibres can slightly reduce the density fluctuation and result in a more uniform profile decreasing the unwanted effect of the surface layer.

### 4.5. Contributions of Energy Components during Deformation

To gain a better insight into the deformation mechanism, in Figure 13, the individual components of energy are tracked as a function of the applied strain which constitutes the bonding, the bond-angle, the dihedral and non-bonded energies. For the sake of comparison, the components are normalized by their initial value. 

As can be seen from Figure 13, the deformation is mainly dominated by the contributions of bond and van der Waals energies. The van der Waals energy increases steadily throughout the deformation. In the elastic phase, the bond energy decreases monotonically as the bond lengths approach the equilibrium value. Upon yielding, the bond energy decreases slightly while the van der Waals term increases steadily, which may be due to the chain slippage mechanism. With further loading, the polymer chains untangle and begin to move away from each other. This changes the bond energy level as the strain in the nanofibre begins to relax, and the slope of the van der Waals curve is also affected at this point. 

The comparison of the energy contributions of the pristine and untreated nanofibres shows that doping with nanoparticles reduces the slope of the van der Waals energy because it restricts the movements of the polymer chains during deformation. On the other hand, in the elastic region, the slope of the bond contribution of energy is increased in the doped nanofibres; this increased stiffness is responsible for the increase in brittleness of the doped nanofibres.

## 5. Summary and Conclusions

In the current work, molecular dynamics simulations are conducted to unravel the effect of doping silver nanoparticles on the mechanical properties of poly(lactic acid) nanofibres. Special attention is devoted to the fracture properties, where the effect of the doping agent on the crack propagation mechanism and energy release rate is studied. The obtained understandings are summarized as follows:The nanoparticles act as consolidators, binding the polymer chains together and preventing them from pulling out. Therefore, they effectively hinder crack propagation and increase the critical energy release rate.Nanoparticles restrain the movement of the polymer chains and reduce the local atomic strain within the nanofibre, thus reducing the shrinkage of the cross-section due to the applied strain and avoiding local necking. This is also reflected in the decrease in the deformability and Poisson’s ratio of doped nanofibres.The brittleness of treated nanofibres is moderately enhanced, reflected in a greater Young’s modulus and yield stress. In addition, the moduli of resilience and toughness reduce in doped nanofibres, also indicating increased brittleness.Comparison of the different weight fractions of doping nanoparticles shows that the results also depend on the position at which the nanoparticles are launched and there may not be a direct relation between the NP weight percentage and the improved fracture properties; however, proper distribution of nanoparticles can ensure increasing the fracture strength of nanofibres. A deeper and more quantitative understanding of the effect of silver nanoparticles can be achieved by considering different variations of weight fraction and decoration.The radial density profile suggests that the presence of doping nanoparticles on the surface of nanofibres can decrease the effect of the low-density surface layer and result in a more uniform density profile.A comparison of the energy contributions of the pristine and untreated nanofibres shows that doping with NPs reduces the change in the van der Waals energy because it restricts the movements of the polymer chains during deformation while they increase the slope of the bond energy in the elastic region which is responsible for the increase in brittleness of the doped nanofibres.

## Figures and Tables

**Figure 1 nanomaterials-13-00989-f001:**
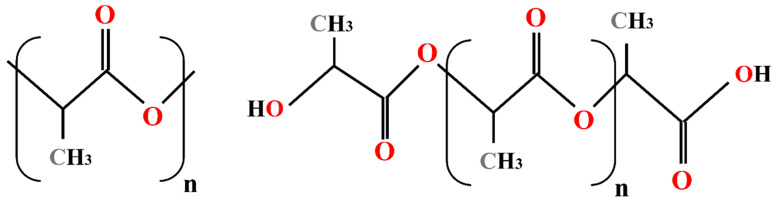
Chemical structure of PLA.

**Figure 2 nanomaterials-13-00989-f002:**
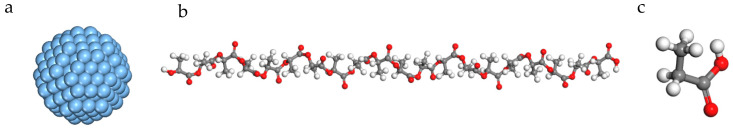
(**a**) Silver nanoparticle. (**b**) A PLA chain consisting of twenty monomers. (**c**) A typical monomer of PLA. The blue, red, grey and white colours refer to silver, oxygen, carbon and hydrogen atoms, respectively.

**Figure 3 nanomaterials-13-00989-f003:**
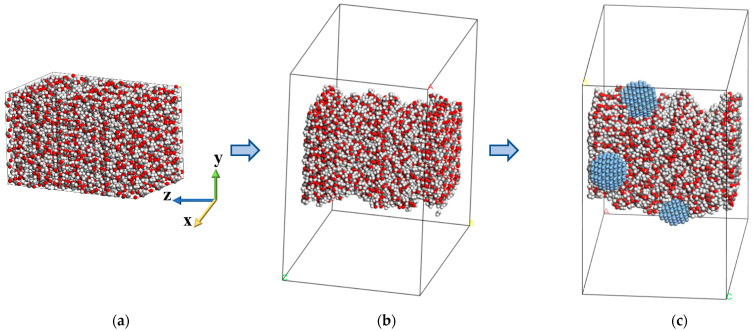
(**a**) Initial cuboid. (**b**) Relaxed nanofibre in the enlarged cell. (**c**) Doping silver nanoparticles on nanofibre.

**Figure 4 nanomaterials-13-00989-f004:**
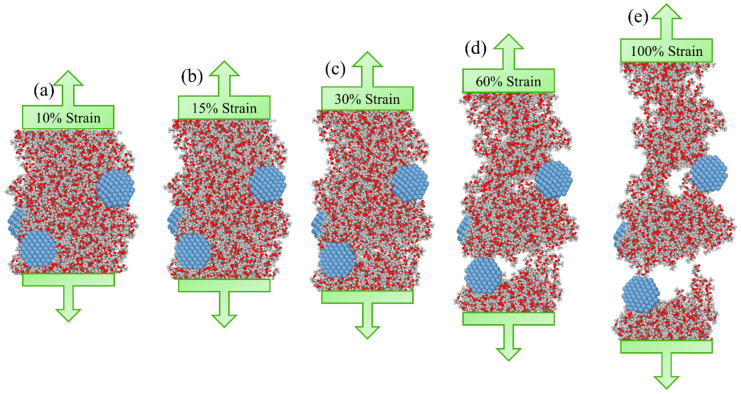
Tensile deformation on the nanofibre doped with 5.8% weight fraction of silver nanoparticles. (**a**) at 10% strain and in the elastic phase (**b**) at 15% strain and in the strain-softening phase due to the chain slippage (**c**) at 30% strain slightly after crack initiation (**d**) at 60% strain and the crack propagation phase due to pulling out of polymer chains (**e**) at 100% strain and close to complete rapture.

**Figure 5 nanomaterials-13-00989-f005:**
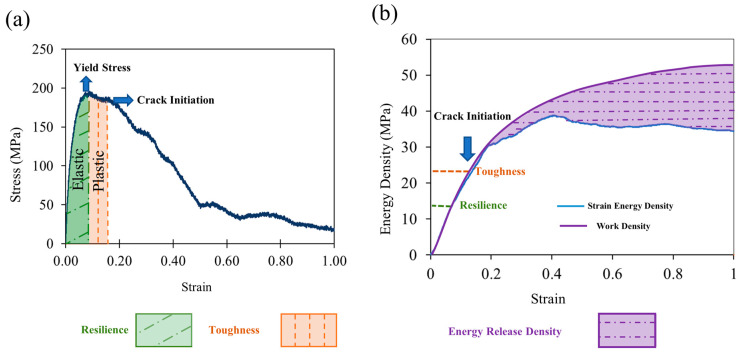
(**a**) Stress–strain response of silver doped PLA nanofibre diameter deformed in uniaxial tension. (**b**) Evolution of strain energy and work densities.

**Figure 6 nanomaterials-13-00989-f006:**
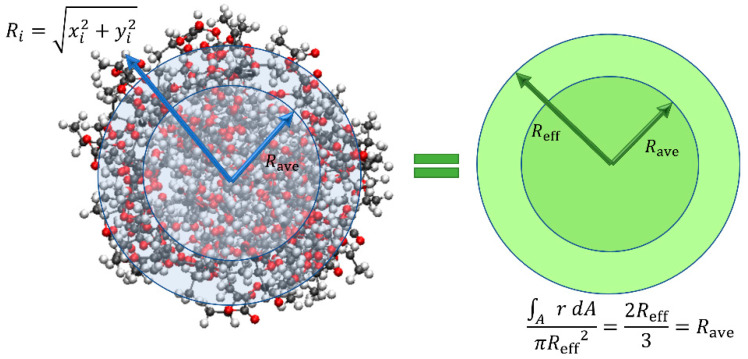
Determination of effective radius.

**Figure 7 nanomaterials-13-00989-f007:**
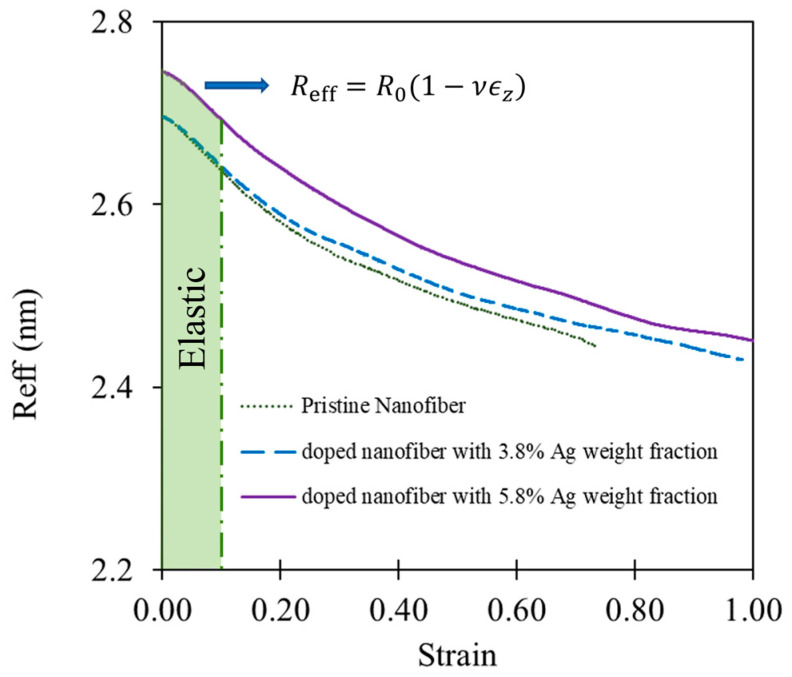
The change in effective radius during deformation and determination of Poisson’s ratio.

**Figure 8 nanomaterials-13-00989-f008:**
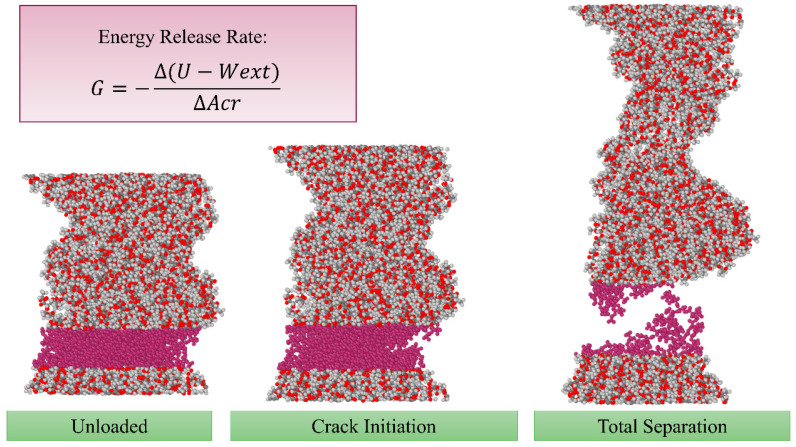
Determination of energy release rate for nanofibres.

**Figure 9 nanomaterials-13-00989-f009:**
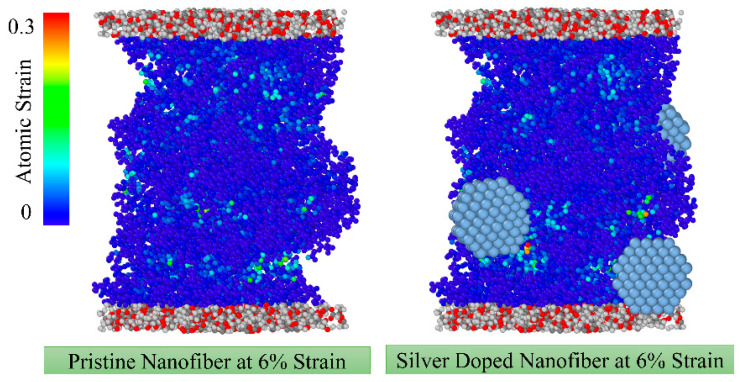
The effect of doping nanoparticles on the distribution of atomic strain in the elastic regime.

**Figure 10 nanomaterials-13-00989-f010:**
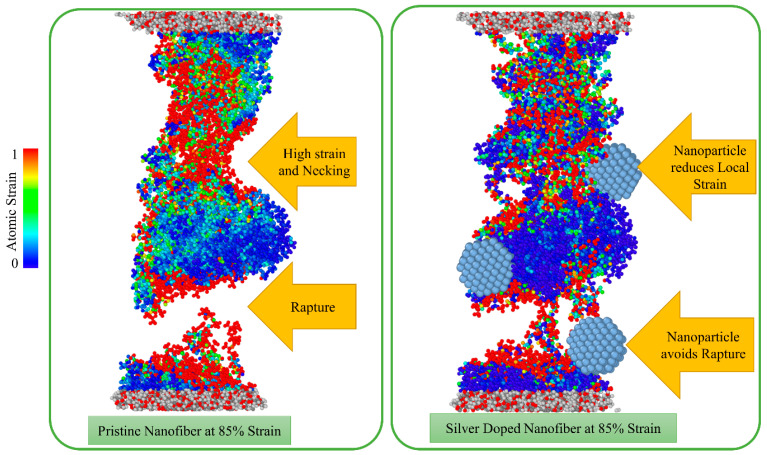
The effective role of doping nanoparticles on fracture resistance of nanofibres.

**Figure 11 nanomaterials-13-00989-f011:**
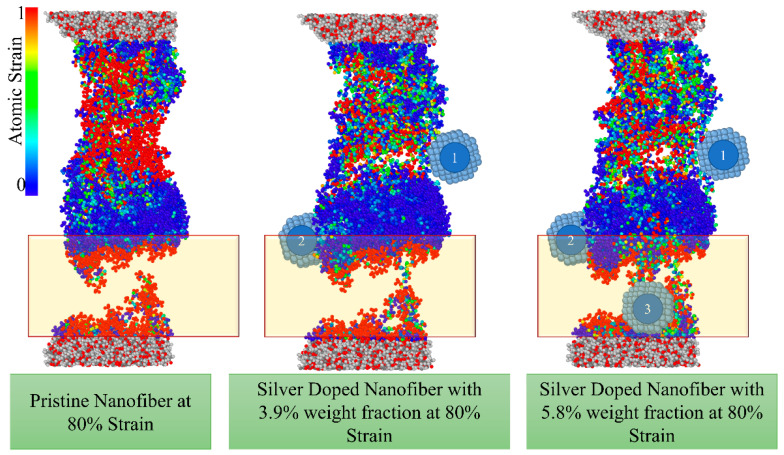
The effect of the weight fraction of nanoparticles on the strain distribution and fracture.

**Figure 12 nanomaterials-13-00989-f012:**
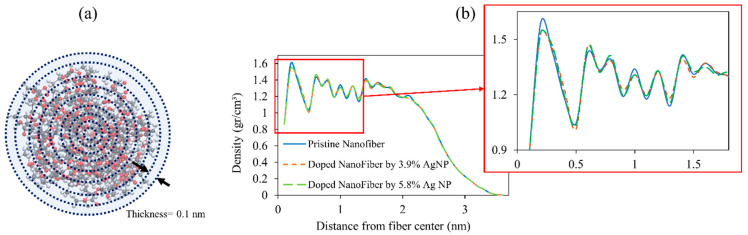
(**a**) The schematic of nanofibre cross-sections divided into concentric cylindrical shells of 0.1 nm thickness. (**b**) Distribution of radial mass density of pristine and doped nanofibres.

**Figure 13 nanomaterials-13-00989-f013:**
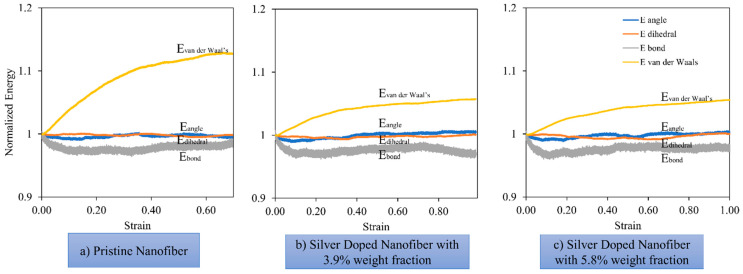
Energy decomposition for pristine (**a**) and doped nanofibres with 3.9% (**b**) and 5.8% (**c**) weight fractions of Ag NP.

**Table 1 nanomaterials-13-00989-t001:** Mechanical parameters of pristine and silver-doped nanofibres from MD simulations.

	Young’s Modulus (GPa)	Poisson’s Ratio	Proportional Stress (MPa)	Yield Stress (MPa)	Resilience (MPa)	Toughness (MPa)	Energy Release Rate (m J/m^2^)
Pristine Nanofibre	2.6	0.22	168	192	15.55	24.85	365
Nanofibre doped with Silver nanoparticle of 3.9% weight fraction	2.8	0.19	178	208	14.80	22.57	496
Nanofibre doped with Silver nanoparticle of 5.8% weight fraction	2.7	0.17	189	193	15.11	23.75	520

## Data Availability

The data presented in this study are available on request.

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
