# Peer review of "A Study on the Effect of Doping Metallic Nanoparticles on Fracture Properties of Polylactic Acid Nanofibres via Molecular Dynamics Simulation"

_nanomaterials, 2023, doi:10.3390/nano13060989_

Round 1
Reviewer 1 Report
The topic of the manuscript is important, but the following points relating to the methodology should be clarified:
1.How do the strain values used correlate with the experiment? Even 10% is a very large deformation, not to mention 100%.
2.What thermostat was used when strain was applied? Asymmetric NPT or other?
3.Please provide reference for Eq.3.
4.It is not clear how exactly the integral below Eq. (3) was calculated and what is dl in the integral.
5. As I understand the electrostatic term was calculated using cutoff. In this method results are very sensible to the cutoff radius. Have you studied the dependence of the results on the cutoff radius?
Author Response
Please kindly find attached the response of Authors to Reviewer's comment.

Reviewer 2 Report
In the current work, molecular dynamics simulations are conducted to unravel the 440 effect of doping silver nanoparticles on the mechanical properties of poly(lactic acid) nan-441 ofibres. Special attention is devoted to the fracture properties, where the effect of the dop-442 ing agent on the crack propagation mechanism and energy release rate is studied. Authors have performend a comprehensive MD simulation study, which can be published after minor revision. I can a concern: only two silver particles in the simulation box. I wonder few silver particles may fail to reproduce the experimental results well, e.g., the effect of doping Ag NP on Fracture properties of nanofibers.
Author Response

(The authors gave the same response as above.)

Round 2
Reviewer 1 Report
"Larger cut-off radius can increase the accuracy of the results [7], however, a study on the amount of sensitivity of the results to the cut-off radius is not conducted in the current work."
Why investigation was performed in [7] and not in the present paper? It should be clarified
Author Response
Please kindly see the attachment for the response to Reviewer's comment.
